

# Windstorm damage relations - Assessment of storm damage functions in complex terrain

Ashbin Jaison[1,2], Asgeir Sorteberg[1,2], Clio Michel[1,2,3], and Øyvind Breivik[1,2,3]

[1]Geophysical Institute, University of Bergen, Bergen, Norway
[2]Bjerknes Centre for Climate Research, Bergen, Norway
[3]Norwegian Meteorological Institute, Bergen, Norway

**Correspondence:** Ashbin Jaison (aja052@uib.no)

**Abstract.** Extreme winds are by far the largest contributor to Norway's insurance claims related to natural hazards. The predictive skills of four different damage functions are assessed for Norway at the municipality and national levels on daily and annual temporal scales using municipality level insurance data and the high-resolution Norwegian reanalysis (NORA3) wind speed data for the period 1985-2020. Special attention is given to extreme damaging events and occurrence probabilities

of wind speed induced damages. Because of the complex topography of Norway and the resulting high heterogeneity of the population density, the wind speed is weighted with population. The largest per-capita losses and severe damages occur most frequently in the western municipalities of Norway whilst there are seldom any large losses further inland. The good agreement between the observed and estimated losses at municipality and national levels suggests that the damage functions used in this study are well suited for estimating severe wind storm-induced damages. Furthermore, the damage functions are able to

successfully reconstruct the geographical pattern of losses caused by extreme windstorms with a high degree of correlation. From event occurrence probabilities, the present study devises a damage classifier that distinguishes between daily damaging and non-damaging events at the municipality level. While large loss events are well captured, the skewness and zero-inflation of the loss data greatly reduces the quality of both the damage functions and the classifier for moderate and weak loss events.

## 1 Introduction

Wind-related damage claims account for 56% of Norway's insurance payouts related to natural hazards from 1980 to 2017 and are by far the largest component of loss claims related to natural hazards (DSB Norway, 2019). However, a detailed investigation into the relationship between Norwegian windstorms and damage has so far not been conducted. The comparison of different proposed storm damage models has only been conducted in a few countries due to a lack of long and sufficiently homogeneous insurance claims data (Cole et al., 2010; Prahl et al., 2015). In this paper, we investigate the relations between

wind storms and their associated damage by analysing 36 years of daily insurance data on the municipality level and daily maximum wind speed data using a set of storm-damage functions.

Establishing robust windstorm-damage relations may help predict storm damage risk using weather forecasting (Merz et al., 2020) and help in evaluating the risk change on the longer term in conjunction with climate change. Moreover, understanding the monetary risk involved in extreme wind events is crucial from the insurer's perspective to fix reasonable premiums and





estimate portfolio risk. Storm-damage functions describe the mathematical relation between the intensity of a natural hazard, here the wind speed, and the monetary loss incurred due to the event. There are mainly two types of storm-damage functions: 1) the storm-based approach, which uses historical losses and wind speed information and 2) the hazard-based approach, which in addition makes use of micro-scale information such as the vulnerability, exposure and value of the assets. However, as detailed information about the damage is often proprietary, the most common approach, also used in the present study, is

the storm-based approach, where the storm hazard is directly linked to the loss information through a storm damage function (Dorland et al., 1999; Klawa and Ulbrich, 2003; Prahl et al., 2012, 2015). These functions can also be split into deterministic and probabilistic types (Prahl et al., 2012). The deterministic damage functions do not estimate the uncertainty in the wind speed-loss relation, whereas the probabilistic damage functions assume a statistical distribution for the model error. To make the deterministic and probabilistic models comparable, estimates from deterministic models are treated as equivalent to the

mean of the estimates from the probabilistic models.

Storm-damage functions must be regionally fitted because they are highly dependent on local features. The amount and size of the damages caused by extreme wind strongly depends on the exposure level of assets (Cardona et al., 2012), which is connected to demography and economy which can change over time due to a variety of reasons such as urbanisation, higher infrastructure standards, economic growth, etc. Moreover, building types, building codes, differing insurance policies, claims

settlement practices can also influence the performance of storm-damage functions (Walker, 2011) especially if the fitted storm-damage functions are applied to different countries. Norway has a complex topography with mountains and valleys and a rugged coastline with many fjords with a major share of the population living along the coasts and in the valleys (Simensen et al., 2021). Therefore, the population density is an important factor to take into account when establishing storm-damage functions (Donat et al., 2011).

Windstorms are most frequently associated with extratropical cyclones whose number may change in the future (Priestley and Catto, 2022). Wind speeds associated with such systems are aggregated over the northern hemisphere is predicted to increase (Priestley and Catto, 2022) hence increasing the potential for damages in addition to the expected increase of the assets amount (Schwierz et al., 2010; Held et al., 2013). Moreover, return values of daily maximum wind speed over Scandinavia are expected to increase meaning more frequent strong winds (e.g., Outten and Sobolowski, 2021). However, many studies have

pointed out the large uncertainty in the wind speed projections on regional and local scales (Outten and Sobolowski, 2021; Christensen et al., 2022; Michel and Sorteberg, 2023).

A number of studies have investigated storm damage and risk on residential structures and other insured losses, mainly for Europe and more particularly Germany, using various damage functions and local information. Dorland et al. (1999) suggested a deterministic damage function by which loss increases exponentially with wind speed such that a slight increase in storm

intensity can cause a significant increase in economic damage in Netherlands and northwest Europe. Meanwhile, analysing annual insurance loss due to windstorms in Germany, Klawa and Ulbrich (2003) advocated a cubic relationship between the deviation in wind speed from its 98th percentile and the loss. Heneka and Ruck (2008) and Heneka and Hofherr (2011) applied a probabilistic damage function for Germany, which incorporates extreme value theory in conjunction with a non-linear function. However, this probabilistic damage function requires both claim and loss ratios data that are not common shared data





and which we lack for Norway. To estimate the daily and annual losses at the district levels in Germany, Prahl et al. (2012)
proposed a power law-based probabilistic damage function where loss is proportional to a power of wind speed and found
out that the exponents range between 8 and 12. Welker et al. (2016) simulated the spatial pattern of losses associated with
historical windstorms that happened in Switzerland using the asset amount and the vulnerability, the latter depending on the
wind gust. The agreement between the simulated loss and the observed insurance loss was shown to be reasonable but also

case-dependent. They pointed to the uncertainty in the input data, such as in the wind gust but also in the estimation of the
assets and vulnerability. More recently, Koks et al. (2020) developed an open-source hazard-based model that uses publicly
available hazard, exposure and vulnerability data and the loss estimates can be treated as a baseline for further research. Using
three different methods, Held et al. (2013) found a steady increase in the values associated with a 10-yr return loss by the end
of the 21st century considering only the German private houses. Schwierz et al. (2010) suggests that, with climate change and

increased storm intensity, Norway can expect a 16% increase in annual losses associated with windstorms. However a recent
study by Severino et al. (2023) indicates a significant decrease in winter storm damage over Norway.

In the following section, we introduce the insurance loss data and NORA3 hindcast wind speed data along with the different
storm damage functions. In section 3, the climatology of the extreme winds and damages is presented in addition to the
modelling results. We summarise and discuss the results in section 4.

## 2   Data and methods

### 2.1   Insurance loss data

We use daily insurance loss data, composed of the daily accumulated number of claims and monetary loss, from the Norwegian
Natural Perils Pool for each of the 356 municipalities constituting Norway. The data span 36 years, from 1985 to 2020. The
loss data distinguish losses by natural event types, such as floods, landslides, storm surges and windstorms. The present study

focuses on the damages associated with windstorms.

Natural peril insurance is a compulsory part of the fire insurance held by almost all households in Norway (Sandberg et al.,
2020). By the Norwegian Natural Perils Pool act, all buildings and movable properties which are insured against fire damage
are also insured against natural disasters. All insurance companies underwriting fire insurance in Norway are obliged to become
members of the Norwegian Natural Perils Pool and archive their losses. The fraction of households having fire insurance has

stayed relatively constant over the period of interest; thus, the effect of varying market penetrations is small. In many previous
studies, loss ratio and claim ratio, which are dimensionless, are used to model storm damage relations (Huang et al., 2001;
Held et al., 2013; Prahl et al., 2015; Welker et al., 2016). However, Norwegian insurance does not include the total insured
value, which prevents us from using the loss and claim ratios in the present study.

For such a long time series of loss data, it is necessary to account for inflation, and economic changes. To counteract the

effect of inflation, the insurance loss is adjusted for inflation using the yearly consumer price index with 2015 as the base year
(SSB Norway, 2023a). As an example, the relative difference of loss after inflation adjustment is of +60% for the New Year
Storm (1992) and +7% Dagmar (2011).
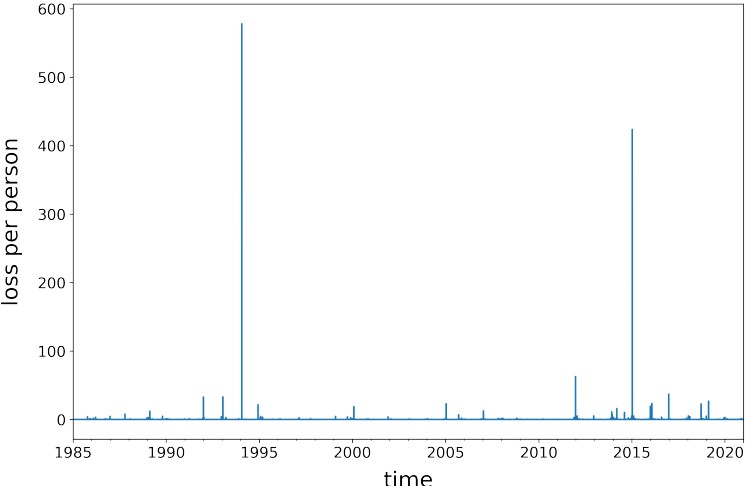

**Figure 1.** Example of time series of daily insurance loss per person (NOK, 2015 values) for an arbitrary municipality.

Changes in exposure is another key variable that determines the intensity of losses incurred. Many studies use population as a proxy for exposure (Simpson et al., 2014). To address the change in exposure, we compute the loss per person for each municipality using the yearly population data given at the municipality level (SSB Norway, 2023b). Other factors, which may influence exposure, such as changing building standards and wealth distribution are not accounted for in the present study.

A few extreme events have caused the majority of the total damage associated with wind storms, with the five largest events causing 4.3 bn. NOK of damages (2015 values), which represents 36% of the total insurance loss. The top damaging events and their associated losses are given in Table 1. As expected from the more frequent cyclones reaching Norway in winter than summer (Hoskins and Hodges, 2019), most extreme events occur between November and April. The presence of such extreme events brings skewness in the loss distribution and the absence of losses on most days of the year makes it zero-inflated. The distribution skewness and the zero inflation (Fig. 1) in loss data restrict us from using conventional fitting methods. Figure 1 highlights a record high number of claims in years 1992, 2011 and 2015. This can be attributed to the New Year Storm in 1992, storm Dagmar in 2011 and storms Nina and Ole in 2015 (Table 1).

## 2.2 The wind speed data: The NORA3 hindcast

The high-resolution hindcast NORA3 covers the period 1979-2021 (and is being extended). The spatial resolution of NORA3 is 3×3 km, and surface variables are archived at hourly resolution. The non-hydrostatic, convection-permitting model HARMONIE-AROME (Seity et al., 2011; Haakenstad et al., 2021; Haakenstad and Breivik, 2022) ingests surface observations through a simple surface analysis scheme and blends this with initial fields from ERA5 (Hersbach et al., 2020). Boundary conditions are also taken from ERA5. The data is publicly available on the website https://thredds.met.no/thredds/catalog/nora3/catalog.html (last access 24 April 2022). The domain covers the North Sea, the Norwegian Sea, the Barents Sea, Svalbard and is bounded





by Finland to the east. The hindcast consists of a sequence of overlapping 9-hourly forecasts (initialized at 00, 06, 12 and 18 UTC). Only time steps 4-9 h are used here. In the present study, we use the hourly wind speed and wind gust from 1985 to 2020, which were the 36 years available at the time of our analysis. The daily maximum near-surface wind speed and gust were

extracted from the 24 hourly wind fields of each day. NORA3 only slightly underestimates the maximum observed wind speed (Haakenstad et al., 2021; Solbrekke et al., 2021) and its interquartile range for the 10 strongest windstorms that affected Norway between 2009 and 2018 (Haakenstad et al., 2021), outperforming both the earlier hydrostatic 10-km Norwegian Hindcast Archive (NORA10, Reistad et al., 2011) as well as the recent ERA5 reanalysis.

## 2.3    Municipality level wind speeds

As the insurance loss is at the municipality level, we must estimate a municipality-relevant wind speed in order to apply the storm-damage functions. In terms of administrative boundaries, Norway is divided into 356 municipalities and 11 counties. We first re-grid the $1 \times 1$ km$^2$ population data published yearly by Statistics Norway (Strand and Bloch, 2009) to the same resolution as NORA3. We assign each population grid cell to the nearest NORA3 grid cell. If more than one non-zero population grid cell

corresponds to a NORA3 cell, we assign the sum of the population grid cells to the NORA3 grid cell. Using the daily maximum wind speed and the population as weights, we obtain daily population-weighted average wind speeds in each municipality. We repeat the process for the daily maximum wind gusts.

## 2.4    Storm-loss models

Storm-damage functions connect the intensity of a storm event to the monetary damage caused by the storm. With the available

historical data of insurance loss and wind speed, we apply the storm-based approach to model storm-damage functions. The storm-damage functions discussed here are macroscale statistical models calibrated at the municipality level. Our key objective is to compare and assess the quality of various proposed storm-damage functions applied to our data. We employ three damage functions: the deterministic exponential model (Huang et al., 2001; Murnane and Elsner, 2012), the deterministic model of Klawa and Ulbrich (2003) and the probabilistic function by Prahl et al. (2012). In addition, we suggest a modified version of

the Prahl model to better simulate the very steep damage curves found in some Norwegian municipalities. All damage models are fitted to loss per person to ensure uniformity among the storm damage approaches and easier inter-comparison of models and parameters. Finally, we devise a simple ensemble mean of the estimates from the four damage functions listed above. In the following, we describe in detail the damage functions applied. From now on, $L$ refers to the insurance loss, $\nu$ to the weighted wind speeds and $d$ to the damage function.

To fit and assess the skill of the storm damage models, we split the data into a testing and a training set. We assign the years from 1985-1989 and 2010-2012 to the testing part. The rest of the data from 1990-2009 and 2013-2020 is the training data. A necessary condition for splitting the data is that training and testing data should have identical distributions. We split the data so that both testing and training data include extreme storm events, the storm Dagmar in the testing data, and the New Year storm in training data.


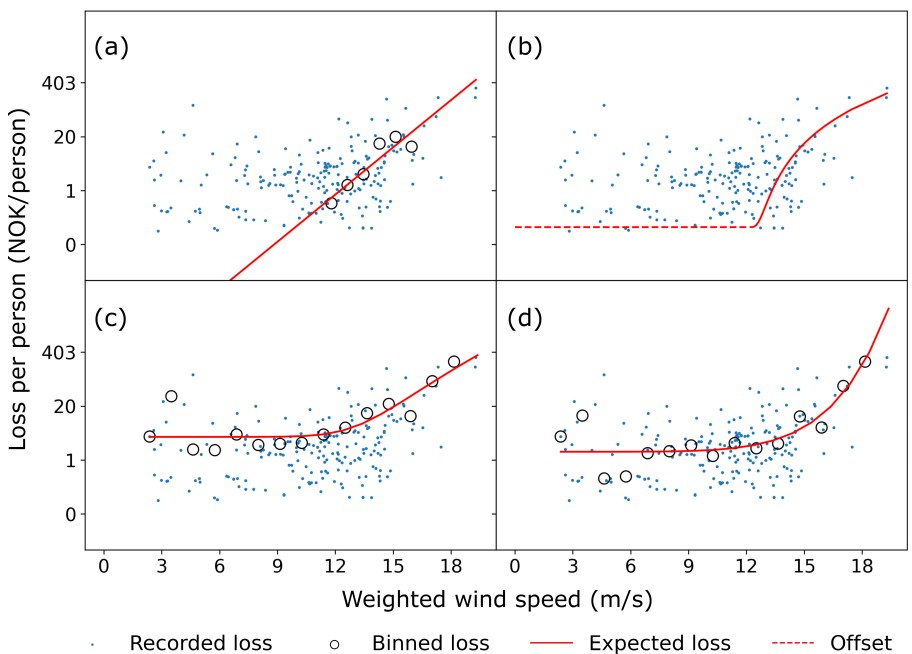

**Figure 2.** Shapes of the damage functions for an arbitrary municipality for (a) the exponential damage function, (b) the cubic excess over threshold damage function, (c) the magnitude term in the probabilistic damage function by Prahl, and (d) the magnitude term in the modified Prahl probabilistic damage function. Note that the y-axis is on a logarithmic scale.

### 2.4.1 Exponential model

The exponential damage function assumes that loss increases exponentially with increasing wind speed (Dorland et al., 1999). It is a simple damage function with only two parameters to be estimated. The damage function is formulated as,

$$d(\nu) = e^{\alpha(\nu - \beta)} \tag{1}$$

where $\alpha$ is the scale parameter and $\beta$ is the location parameter. Then the loss is estimated from the damage function as $L(\nu) \propto d(\nu)$. Due to the random nature of losses for lower wind speeds, we fit the damage function for wind speeds above the 95th percentile. The associated loss values are split into ten equally spaced bins with respect to the wind speeds and with a pre-condition that at least five loss days belong to each bin. The binned losses are log-transformed, and with the assumption of normality, least square methods are employed to estimate the model parameters. Figure 2a shows the shape of the damage function with the red line.





### 2.4.2 Cubic-excess over threshold model

The damage function proposed by Klawa and Ulbrich (2003) suggests that the loss increases cubically for wind speeds beyond a certain threshold. More precisely, the damage function is the third power of wind speeds above the 98th percentile scaled by the 98th of the percentile wind speed:

$$d(\nu) = \left( \frac{\nu - \nu_{98}}{\nu_{98}} \right)^3 \tag{2}$$

The loss is obtained by linear regressing the damage function (Eq. 3):

$$L(\nu) = \beta_0 + \beta_1 d(\nu) \tag{3}$$

The intercept term $\beta_0$ in the fitted linear regression can be interpreted as the base loss, which is the loss estimate for all wind speeds below the 98th percentile. However, using this loss offset for all wind speeds below the 98th percentile doesn't allow to address the randomness in the lower loss spectrum. Figure 2b shows the model fit for this damage function (see the red solid and dashed lines).

### 2.4.3 Probabilistic damage function by Prahl

The power law based probabilistic damage function by Prahl et al. (2012) consists of a two-step fitting procedure, the first step estimating the occurrence probability of damage for a given wind speed and the second step estimating the loss magnitude. The loss data is binned with respect to wind speed and we calculate the occurrence ratio (i.e. occurrence probabilities) in each bin. We then fit the following sigmoid function to the binned wind speeds.

$$p(\nu) = 1 - \frac{\gamma_0}{1 + e^{\gamma_1(\nu - \gamma_2)}} \tag{4}$$

where the parameter $\gamma_1$ determines the steepness of the curve, $\gamma_2$ separates the wind speed threshold beyond which the curve gets steeper and $\gamma_0$ determines the base probability of losses. Figure 3 shows the fit of the probability term (Eq. 4) of the damage function (see the red line). In addition, for a given wind speed $\nu$, the magnitude of the loss M for non-zero losses is estimated through a power law-based function (Fig. 2c) and is related to the wind speed as follows,

$$M(\nu) = \sigma_0 + \left( \frac{\nu}{\sigma_2} \right)^{\sigma_1} \tag{5}$$

where $\sigma_2$ scales the wind speed, $\sigma_1$ is the shape parameter and $\sigma_0$ is the offset loss. The magnitude term is fitted on losses binned with respect to wind speeds.

The probability term makes use of the whole loss range while the magnitude term only uses non-zero losses. The probability of damage and the magnitude of loss are treated as independent variables. The damage function is then the product of the


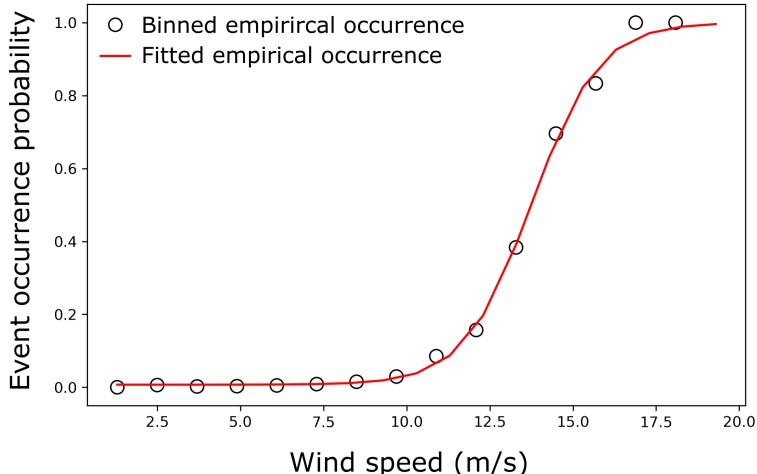

**Figure 3.** Example of sigmoid function that estimates the probability of an event occurrence for an arbitrary municipality. The estimated parameters in this municipality are: $\gamma_0$=0.99, $\gamma_1$=0.99 and $\gamma_2$=13.75.

probability and the magnitude of loss:

$$d(\nu) = p(\nu)M(\nu) \tag{6}$$

The damage function includes the assumption that the observed losses follow a log-normal distribution ($M_{obs} \sim \mathrm{LN}(\mu, \sigma)$, where $M_{obs}$ is the observed non-zero loss). Therefore, the expected loss for a given wind speed is

$$L = p(\nu)E(M(\nu)) \tag{7}$$

The probabilistic damage function by Prahl has a complex fitting procedure with eight parameters to be estimated. We refer the readers to the work by Prahl et al. (2015) to learn more about the parameters and fitting procedures of the model. The location ($\mu$) and scale parameters ($\sigma$) of the log-normal distribution are estimated using the maximum likelihood method, and the other parameters of the damage function are estimated with the least squares method.

### 2.4.4 Modified probabilistic damage function by Prahl

The rationale behind Prahl's damage function is that the loss increases steeply for extreme wind events (Fig. 2c). Based on empirical evidence, there is a need for an even steeper damage function for certain municipalities in Norway. Therefore, we propose a modified version of the damage function by Prahl. The magnitude term in Eq. (5) of the Prahl damage function is modified as follows:

$$M(\nu) = \sigma_0' \exp\left[ \left( \frac{\nu}{\sigma_2'} \right)^{\sigma_1'} \right] \tag{8}$$

The rest of the fitting procedure and assumptions are the same as for the Prahl damage function. The shape of the magnitude term in Eq. (8) is displayed in Fig. 2d with the red line.



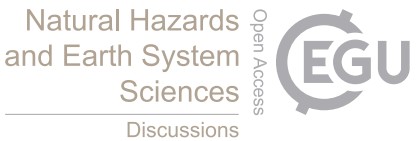

### 2.4.5 Ensemble mean method

The four damage functions presented above have different advantages and drawbacks. The ensemble mean is calculated as the
arithmetic mean of the loss estimates of the four functions, in the hope to improve the overall accuracy.

## 2.5 Model evaluation metrics

We evaluate the models' performance on the training and testing parts at the municipality level using the mean absolute error (MAE) and coefficient of variation (CV). In addition, the predictive skill of the probabilistic function in the Prahl damage function is evaluated using accuracy, recall, precision and F-scores. For robust storm-damage relations, extreme care should
be taken in the parameters estimation of damage functions. To ensure robustness of the damage functions, we bin the loss data with respect to wind speeds to eliminate the sensitivity of damage functions to extreme events.

As its name indicates, MAE is the mean of absolute differences between the observations and the model fits and is formulated as:

$$\text{MAE} = \frac{1}{n} \sum_{i=1}^{n} |y_i - \hat{y}_i| \tag{9}$$

where $y_i$ is observed loss and $\hat{y}_i$ is the estimated loss. Another evaluation metric used here is CV based on the root mean square error defined by Prahl et al. (2015) as follows,

$$\text{CV} = \frac{1}{\bar{y}} \left( \frac{1}{n} \sum_{i=1}^{n} (y_i - \hat{y}_i)^2 \right)^{\frac{1}{2}} \tag{10}$$

where $\bar{y}$ is the mean of the observed loss. High values of CV indicates large loss variability compared to the mean loss.

To quantify the classification accuracy of the damage classifier, we employ the precision, recall and F-scores, which are
defined as follows:

– Precision: The proportion of correctly classified positive samples to the total number of samples classified as positive.

– Recall: The proportion of correctly classified positive samples to the total number of positive samples.

– F-Score: It indicates the balance between precision and recall. Theoretically, the F-score is defined as the harmonic mean of precision and recall. F-score ranges between 0 and 1 and the higher the F-score the better. We take advantage of the
F-scores to decide the probability threshold for the damage classifier.

The binary damage classifier is optimized using the precision-recall curve and associated F-scores. The F-scores evaluate the ability of a classifier to minimize false positives and capture true positives simultaneously. The probability threshold at which the F-score is maximum is chosen as the split point for the event classifier.


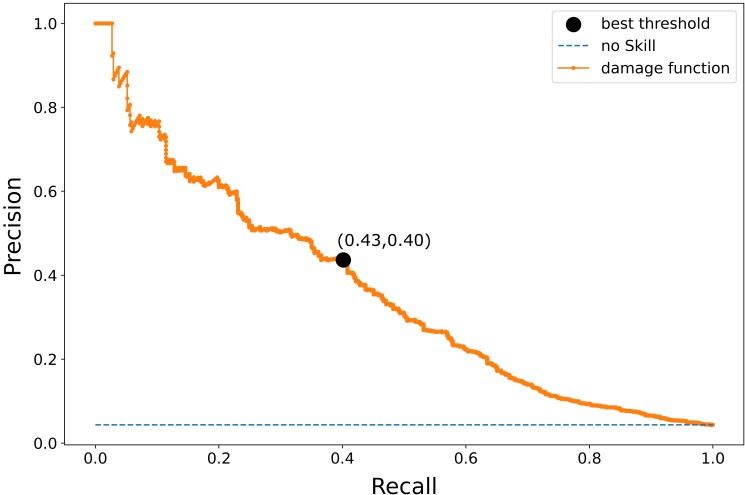

**Figure 4.** The precision-recall curve for an arbitrary municipality is shown with the orange line, the dashed blue line is the model with no skill, and the black dot corresponds to the point where the F-Score is maximum. In this example, the highest F-score of 0.41 is achieved at the probability threshold of 0.30. The precision and recall are shown in brackets.

## 2.6 Damage classifier

A damage classifier labels a given wind speed as damaging or not and adds useful information for event preparedness. The probabilistic damage occurrence probability function in Eq. (4) gives us the opportunity to define a classifier that distinguishes between a damaging and a non-damaging event. To build a robust classifier, it is necessary to define the probability threshold that separates an event from a non-event. With non-event days outnumbering the event days (class imbalance), it is not straightforward to define the probability threshold as 0.5 or to evaluate the model performance for various probability thresholds on

the basis of traditional receiver operating characteristic (ROC) curves and the corresponding area under the curves (AUC). To circumvent the problem of class imbalance in identifying the best probability threshold, we employ the precision-recall curve and the associated F-scores. Figure 4 shows the precision-recall curve for an arbitrary municipality. The split point of damage classifier corresponds to the probability threshold with the highest F-score from the precision-recall curve.

## 3 Results

We analyse the spatial and temporal spread of the insurance loss and compare the population weighted daily maximum wind speed, population weighted daily maximum wind gust and the daily maximum wind speed at the municipality level. We then compare the different damage functions along with the modified Prahl and ensemble mean models.

Considering the high degree of detail involved, we emphasize the following results, (1) daily losses at the municipality level, (2) top three extreme damaging wind events during the study period, (3) losses aggregated to the national level and (4) the

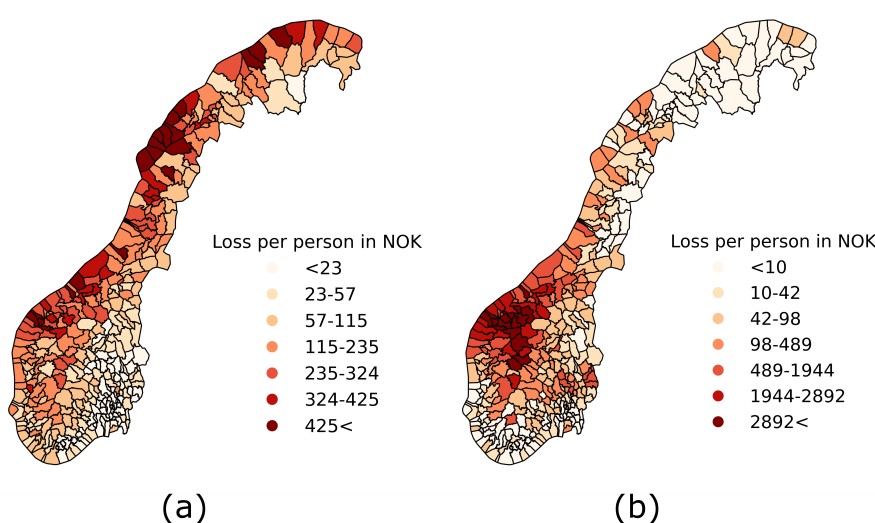

**Figure 5.** Spatial maps of Norway showing (a) the annual average loss per person in each municipality during the study period 1985-2020 and (b) the average loss per person in each municipality caused by the storm Dagmar (25.12.2011-27.12.2011). To account for the skewness involved in the loss data and for meaningful visualization, the loss is split non-linearly with class boundaries at the 20th, 40th, 60th, 80th, 90th and 95th percentiles.

probability function in Prahl et al. (2012) as a classifier. Furthermore, we discuss the pitfalls of the loss data, wind data and storm damage functions.

### 3.1 Overview of the windstorms losses

The municipalities on the west coast of Norway experience higher losses per person whereas there is hardly any loss further inland in southeastern Norway (Fig. 5a). Skewness and zero inflation are especially high for some municipalities in southeastern 245 Norway where wind-related losses are rare. This data-scarcity greatly limits the performance of the damage functions.

The ten most damaging windstorms, in term of cost, that reached Norway during the study period occurred between October and March and mainly impacted central and southwestern Norway and more marginally Northern Norway (Table 1). An example of such a damaging storm is Dagmar in 2011 that impacted western Norway causing more than one billion NOK of losses (Fig. 5b). The insurance losses caused by the ten largest events are given in Table 1 and represent a total of 5347 million 250 NOK, which is 44% of the total losses due to windstorms between 1985 and 2020.

We find no significant temporal trends in the insurance losses caused by extreme winds. Trends in the losses time series, should arise from inflation or changes in wealth distribution. However, the effect of inflation is nullified by adjusting the insurance losses with the consumer price index and a change in wealth distribution is overlooked by the skewness in the losses. Therefore, the Mann-Kendall trend test we conducted on the annual national losses (Fig. 6) fails to detect for any significant 255 trend in losses.





| Event | Number of claims | Loss in million NOK (2015) | Period | Region of impact |
|---|---|---|---|---|
| New Year storm | 22823 | 1933 | 01.01.1992 | Western Norway |
| Dagmar | 14247 | 1274 | 25.12.2011 - 27.12.2011 | Western Norway |
| Nina | 9525 | 593 | 09.01.2015 - 12.01.2015 | South-west Norway |
| Storm of 1994 | 5306 | 261 | 23.01.1994 | South-west Norway |
| Ole | 2418 | 237 | 07.02.2015 - 10.02.2015 | West and northern Norway |
| Storm of 1987 | 5014 | 235 | 16.10.1987 - 17.10.1987 | Southern Norway |
| Frode | 2876 | 234 | 12.10.1996 - 13.10.1996 | Northern Norway |
| Tor | 3940 | 214 | 29.01.2016 - 31.01.2016 | Western Norway |
| Storm of 1988 | 2853 | 184 | 22.12.1988 - 23.12.1988 | Central western Norway |
| Narve | 2080 | 182 | 17.01.2006 - 24.01.2006 | Northern Norway |

**Table 1.** Features of the most damaging storm events that occurred in Norway during the study period.

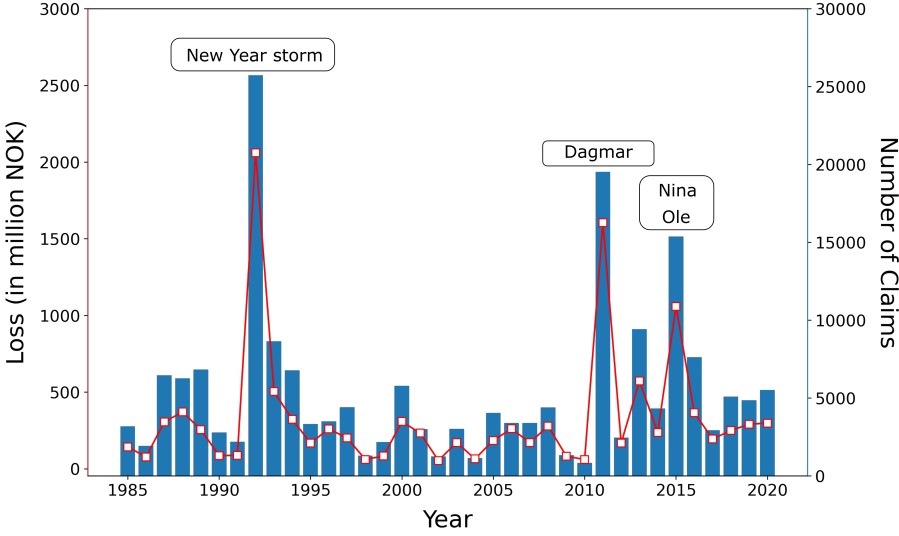

**Figure 6.** Annual number of insurance claims (blue bars) and monetary losses (red line, 2015 inflation-adjusted) associated with strong wind events in Norway from 1985 to 2020.

The choice of wind data has the potential to influence the performance of the damage functions (Prahl et al., 2015). Also, the 98th percentile wind speed is widely regarded as critical from a damage perspective (Klawa and Ulbrich, 2003; Schwierz et al., 2010; Donat et al., 2011). Figure 7a shows that the west coast and Northern Norway experience high magnitude wind events in comparison with south eastern municipalities. The 98th percentile of the population-weighted daily maximum wind speed exhibits a high correlation with the 98th percentile of the population-weighted daily wind gust (0.91) but a lower correlation


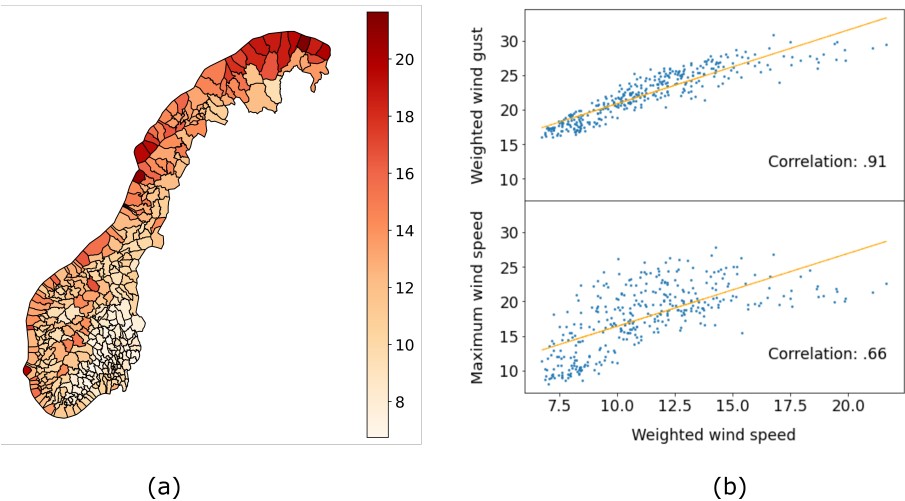

(a)                                                      (b)

**Figure 7.** (a) Map showing the 98th percentile of the population-weighted daily maximum wind speed for each municipality for the period 1985-2020. (b) Spatial correlations between the 98th percentile of the population-weighted daily maximum wind speed and the 98th percentile of population-weighted daily wind gust (top) and the 98th percentile of the daily maximum wind speed (bottom).

with the 98th percentile of the unweighted daily maximum wind speed (0.61). This difference can be attributed to the added information of population as weights for wind speed and emphasizes the importance of accounting for demography (Fig. 7b). From now on we only use the population weighted wind speeds when fitting the damage functions.

From the damage perspective, extreme damaging events are of the topmost concern. The losses above the 99.7th percentile

(occurring on average around one time each year) in each municipality when aggregated account for 85% of total national loss. In each municipality we define the losses higher than the 99.7th percentile as the extreme loss class and losses lying between the 98th and 99.7th percentile as the high loss class. The extreme loss class represents approximately 31 and 9 extreme loss days in each municipality in the training and testing data, respectively.

By applying the different damage functions and calculating the ensemble means, we get daily fits of insurance loss for 10227

days in the training dataset and predictions for 2922 days in the test dataset for every municipality in Norway.

### 3.2 Municipality level loss estimations

The deterministic damage models, Klawa and exponential damage functions perform the best in nearly two thirds of the municipalities across all losses classes in terms of MAE. Table 2 shows the performance of the four different damage functions defined in the methods section and of the ensemble mean for different loss classes. The deterministic models exhibit the

smallest median MAE across all municipalities. Using the CV as the evaluation metric gives similar models performances as when using the MAE. The ensemble mean method does not massively outperform the competing models, but tends to give better results than the Prahl's damage functions (Table 2).


The spatial distribution of MAE is not uniform, but can be linked to the magnitude of the variance of losses, with municipalities with large loss variance having the largest MAEs (Fig. 8). In addition, the spatially heterogeneous distribution of losses
(Fig. 5a) gives rise to spatially heterogeneous errors (Fig. 9a). The CV which shows the extent of variability in losses in relation to the mean losses, is higher in the central and eastern municipalities (Fig. 9b) because the damaging windstorms rarely reach these regions of Norway. On the other hand, the western and coastal municipalities are occasionally hit by wind strong enough to cause damage, which results in a higher mean absolute error in this part of Norway (Fig. 9a).

| Loss class | Damage function | Number of municipalities | MAE | CV |
|---|---|---|---|---|
| All loss days | Modified Prahl | 32 | 31 | 245 |
| | Prahl | 46 | 28 | 238 |
| | Klawa | 134 | 22 | 212 |
| | Exponential | 91 | 24 | 218 |
| | Ensemble mean | 53 | 26 | 226 |
| High loss class | Modified Prahl | 16 | 5 | 184 |
| | Prahl | 25 | 5 | 176 |
| | Klawa | 141 | 4 | 156 |
| | Exponential | 99 | 4 | 134 |
| | Ensemble mean | 39 | 5 | 161 |
| Extreme loss class | Modified Prahl | 53 | 73 | 153 |
| | Prahl | 65 | 67 | 147 |
| | Klawa | 121 | 49 | 132 |
| | Exponential | 75 | 55 | 143 |
| | Ensemble mean | 42 | 66 | 143 |

**Table 2.** Number of municipalities for which a model performs the best, that is has the smallest MAE as a function of the loss class, as defined in the text. The medians of the MAE and CV on all 356 municipalities are also given. Note that the results are based on the performances on the unseen testing data. Also, some municipalities are not evaluated in the high loss class due to lack of data.

## 3.3 Extreme damaging events

The skill of a damage function can be attributed to its ability to reproduce the damages associated with extreme wind events. Moreover from the insurers point of view, extremely damaging wind storm events are of foremost importance. Using only the model exhibiting the best performance in each municipality, we manage to reproduce the spatial pattern of the damages for the three most damaging wind storm events (Fig. 10, see Fig. S1 in supplement for estimates from individual models for the three most damaging wind storm events and Table S2 shows their corresponding correlations. Also, Fig. S2 shows spatial patterns of

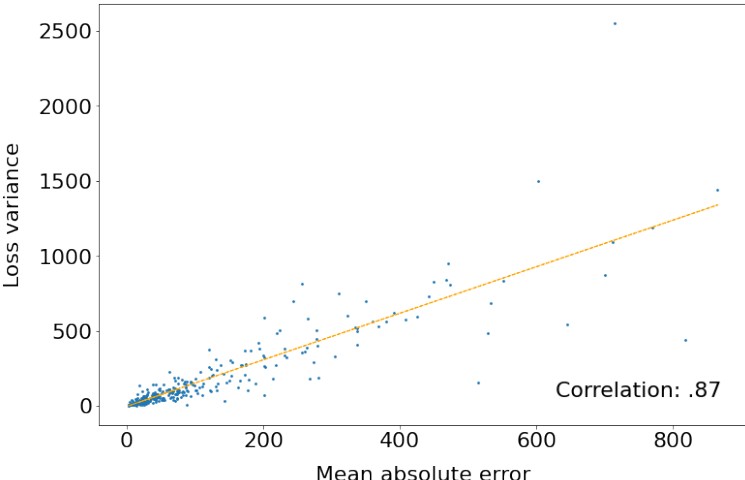

**Figure 8.** Scatter plot of the loss variance against the smallest MAE for losses above the 99.7th percentile in the testing data where each dot represents a municipality. The orange line represents the linear trend obtained using a least squares regression. The spatial correlation is indicated in the bottom right corner.

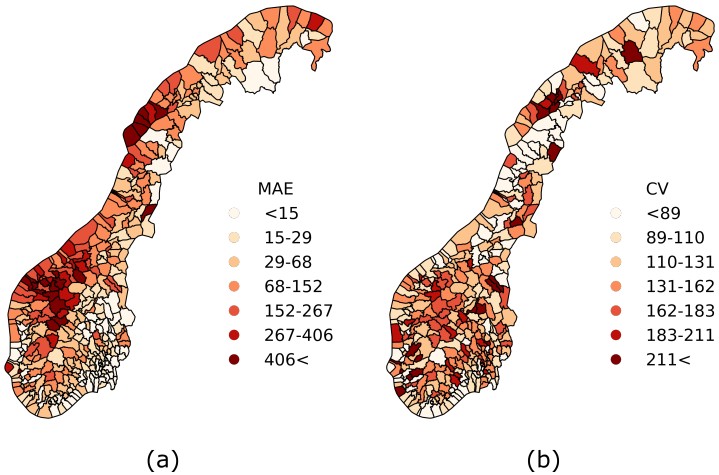

(a)                                        (b)

**Figure 9.** (a) Smallest MAE among the five models in the extreme loss class fitted on the test data and (b) the corresponding coefficient of variation of the root mean square error. The color maps have non-linear class boundaries at the 20th, 40th, 60th, 80th, 90th and 95th percentiles. Note that the results are based on the performances on the unseen testing data.

seven other damaging events.). Statistically significant spatial correlations between the observed and estimated losses reaffirm the suitability of the damage functions in estimating the economic impacts of extreme damaging events.



In the extreme loss class, the probabilistic damage functions and the Klawa damage function perform best in a third of the municipalities each. The Klawa damage function also shows the smallest median error in the extreme loss class which is in agreement with results of previous comparison studies on storm damage functions over Germany (Prahl et al., 2015).

## 3.4 National level loss

The large variance and zero-inflation in the insurance losses at the municipality level decreases when aggregating the data to the national level. Aggregating the municipality level loss observations and estimates yields a time series of daily national loss for each model and we find an overestimation of low-magnitude losses as all damage functions are calibrated in favour of extreme losses (Fig. 11). Moreover, the models' estimates well capture the magnitude and temporal evolution of the observed annual losses at the national level, with a Spearman rank correlation of 0.84 (Fig. 12). Figure 12 also reveals that the losses are slightly overestimated in the training period in years where extreme storm events have occurred while there is an underestimation of loss in 2011 (part of testing data) when storm Dagmar occurred. The aggregated annual national level losses for individual models are shown in Fig. S3 and Fig. S4 in the supplement. Figure S3 shows that the deterministic models are well able to estimate national level losses. The probabilistic models over-estimates the losses in certain municipalities by a large margin reducing the models' ability to estimate national level losses (Fig. S4). While fitting the probabilistic damage functions there are not enough extreme loss observations in certain municipalities which restricts us from being strict on minimum number of loss observations in each bin. This is a reason for huge margin between the observed and estimated losses of the probabilistic models.

## 3.5 Probability of damage occurrence

Predicting an actual damage event using the damage classifier (section 2.6) was rather unsuccessful. The probability thresholds span the whole range of possible values (0.02-0.99) with a median of 0.23. Also, the thresholds for the damage classifier are low in the majority of the municipalities (Fig. 13a) and this is a consequence of the noisy lower loss regime. Southeastern Norway is the region with the highest thresholds because the rarity of extreme damaging wind events (Fig. 13b). With the damage/no-damage thresholds, estimated using the F-score (see section 2.6) we found that the classifier was only able to detect 20-40% of the actual damaging events in most municipalities and zero events in around 15% of the municipalities. This result is due to the large number of days with only a few reported loss, which is exceptionally hard to model (Fig. 14). On the positive side, the damage/no-damage classifier exhibits decent skills in predicting the extreme loss events. On average over 70% of the municipalities have the event occurrence correctly predicted for the top five extreme events (Table 3).

## 4 Conclusions

Wind storms are the natural hazard that makes up more than half of the monetary losses in Norway. The capability of storm damage functions to reproduce the monetary losses associated with damaging wind events are evaluated for the complex topography and demography of Norway. The models' ability to reproduce spatial loss patterns of extreme loss events as


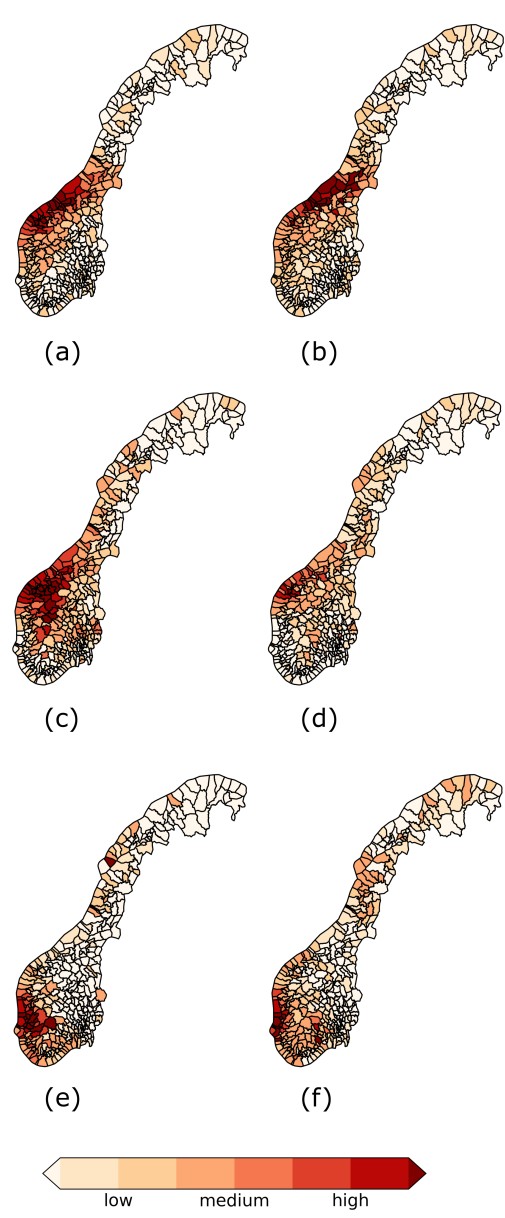

**Figure 10.** Spatial patterns of observed and estimated losses for the three most damaging events, where (a), (c) and (e) display the observed losses of the New Year storm, storm Dagmar and storm Nina, and (b), (d) and (f) are their respective estimates from the closest model to the observed loss. The class boundaries of the colour bar are the 20th, 40th, 60th, 80th, 85th, 90th and 95th percentiles of the observed losses of their respective events. The spatial correlation between observed and estimated losses of the New Year storm, storm Dagmar and storm Nina are 0.67, 0.58 and 0.62 respectively.

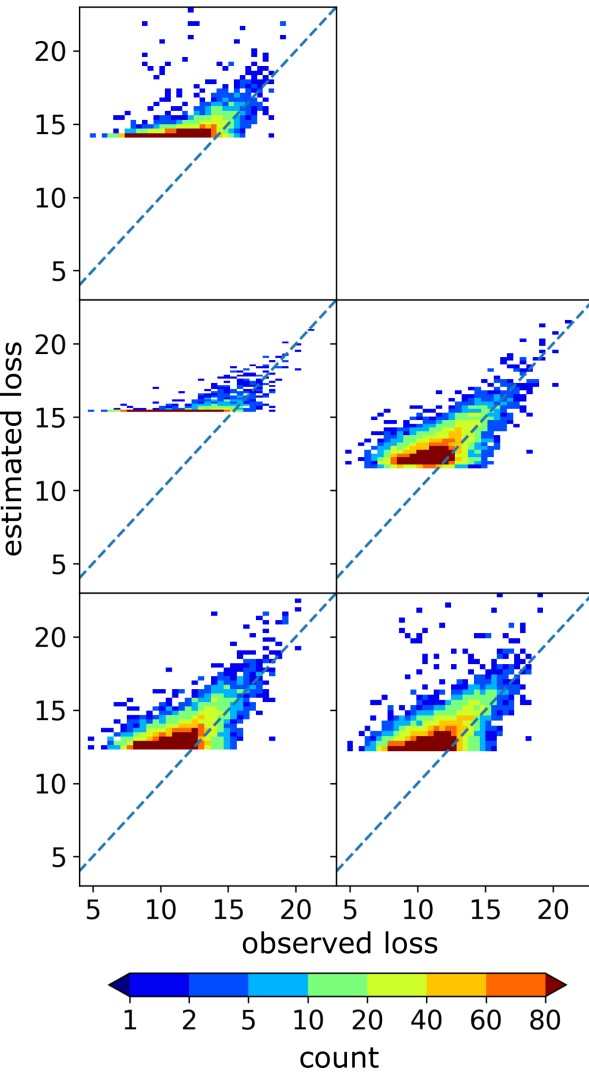

**Figure 11.** Observed and estimated daily losses (NOK) on log-log scale at the national level. Panels (a), (b), (c), (d) and (e) correspond to the ensemble mean method, the Klawa damage function, the exponential damage function, the Prahl damage function and the modified Prahl damage function, respectively. The dashed blue lines represent the 1:1 line.

well as aggregated annual national level losses with a high degree of accuracy confirms the utility of both deterministic and probabilistic damage functions in estimating extreme loss events. However, the relatively poor performance of the damage/no-

325   damage classifier points towards the difficulty of developing an early warning system that encompasses also small loss events. The predictive performances of the damage function and the damage classifier confirms the importance of weighting wind speed with population for better performance of the damage functions.

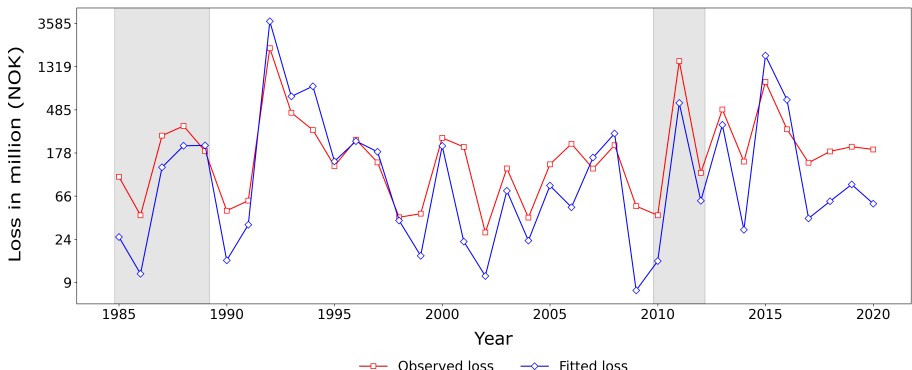

**Figure 12.** Annual time series of observed and estimated national losses using the extreme loss class and the corresponding models for municipalities as given in Table 2. Note that the y-axis is logarithmic and the shaded region represents testing period.

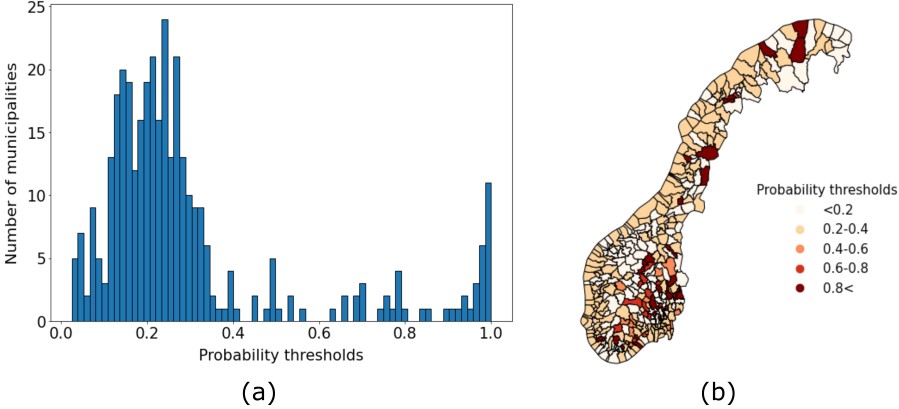

**Figure 13.** (a) Distribution of the probability thresholds for all municipalities and (b) map of the best probability thresholds for each municipality based on precision-recall curves and F-scores.

The deterministic Klawa model performs best in estimating extreme losses and this result is consistent with the previous studies (Prahl et al., 2015). In our study, the Klawa model also exhibits the smallest error in the entire loss range. But, the Klawa model's inability to account for losses below the 98th percentile greatly limits its applicability in the lower loss range. The Prahl damage functions' has the ability to model the whole loss range and shows the smallest error in a third of the municipalities. The models' performances suggest that relying on a single damage model may not be the best strategy if all the municipalities in Norway are to be modelled. Also, the ensemble mean method fail to outperform the individual models. Although the damage/no-damage classifier does very well in predicting extreme damaging events, more research is needed to propose a well-functioning damage classifier across all loss ranges.

There are several limitations to the damage functions including the inability of the models to account for the temporal nature of losses and the sensitivity towards extreme losses. Furthermore, the randomness of losses towards the lower loss spectrum

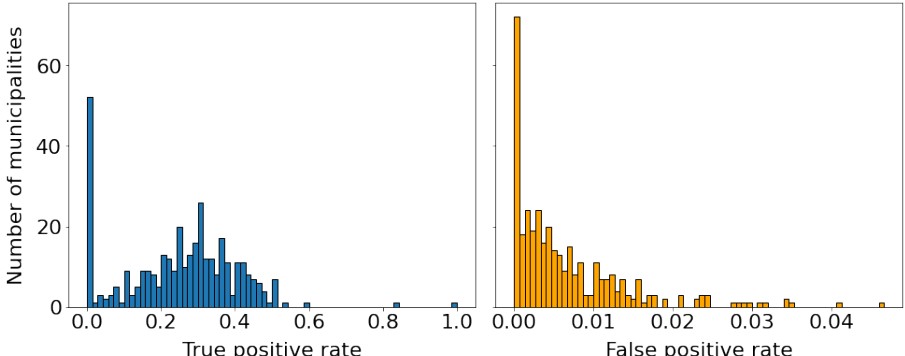

**Figure 14.** Histograms of (left) true and (right) false positive rates of the damage classifier using the testing data.

| Event | Accuracy (%) | Recall (%) |
|---|---|---|
| New Year storm | 75.5 | 77 |
| Dagmar | 66.6 | 63.4 |
| Nina | 76.7 | 71.5 |
| Storm of 1994 | 69.7 | 86.7 |
| Ole | 69.7 | 60 |

**Table 3.** Classification accuracies and recall scores for the top five damaging wind events in Norway ordered with decreasing monetary loss (see Table 1). The accuracy column shows the proportion of municipalities in which the damage classifier accurately predicts both the events and non-events. Recall scores indicate the proportion of municipalities where the damage classifier was able to predict an event that actually occurred.

diminishes the damage classifier's predictive skill. There are also certain pitfalls in the insurance data, such as incorrect report-
ing of time, location and type of claims. Also, the slight underestimation of maximum wind speeds in NORA3 may affect the
340  shape of the damage curves. A direct comparison between other studies that employs damage functions is not possible because
unit of loss in this study is NOK per person while most other studies uses dimensionless loss data.

Studies suggest that with climate change the intensity of future wind storms may increase (Priestley and Catto, 2022). It
would be worthwhile to assess the future changes in wind storm induced losses using the damage functions discussed and
future wind speed projections. Impact based forecasting by which risks associated with a natural hazard are predicted on the
345  short term is gaining more popularity for climate risk management (Taylor et al., 2018; Zhang et al., 2019). The performance
of these damage models, especially on regional level, suggests that it would be possible to train models to forecast damage.
This could be done by adapting the training data developed here for operational forecasts from a numerical weather prediction
model. The training data would have to be corrected for biases specific to NORA3 by comparing archived operational forecasts





in a period overlapping with the NORA3 hindcast. Also, from the risk modeling perspective, coupling the damage functions
with the asset exposure is a possible future direction.

*Author contributions.* AJ: conceptualization, analysis, visualisation, interpretation of results, drafting of the paper. AS: conceptualization, supervision of the work, funding, interpretation of results. CM: post-processing of NORA3 data. ØB: supervision of the work, funding. All authors contributed to drafting and reviewing the manuscript.

*Competing interests.* The authors declare that they have no conflict of interest.

*Acknowledgements.* This work was funded by the Research Council of Norway through the project StormRisk nr 300608 granted to A. Sorteberg. We also thank the Norwegian Natural Perils Pool and the Norwegian Meteorological Institute for providing the data.



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
