# Peer review of "Windstorm damage relations - Assessment of storm damage functions in complex terrain"

_Natural Hazards and Earth System Sciences, 2023_

## Author Comment (AC1)

**Reply to referee comment 1**

**Windstorm damage relations - Assessment of storm damage functions in complex terrain**

We thank the reviewer for reading our manuscript and giving thoughtful comments and suggestions. A detailed response to all comments is found below in blue.

The paper investigates the skill of four storm damage functions to reproduce loss from windstorm events for the complex topography of Norway at both municipality and national level. The damage functions use insurance data and the high-resolution reanalysis dataset NORA3 for the period 1985-2020. The authors show that all damage functions are able to reproduce extreme loss events.

The paper covers an interesting and relevant topic that could be of interest to *NHESS* readers. However, major concerns regarding the data and the adopted methods (see main points below) need to be addressed before considering the paper suitable for publication. Since these concerns are rather substantial, I suggest rejecting the manuscript at this stage, but would encourage the authors to resubmit it once it has been revised.

**Main points**

**GENERAL**

**Comment 1**

Are your results representative/transferable to other regions with complex orography? If not, you should mention Norway in the title.

We have not investigated to what extent the results are transferable to other regions and the original title referred to the selection of wind speeds in order to obtain representative values for damage modelling. We appreciate the comment by the reviewer and have changed the title to:

*Assessment of windstorm-damage relations in the complex terrain of Norway*

**Comment 2**

The study needs to be motivated more clearly in the introduction

In the first paragraph of the introduction we have added the following lines (in italics) on the impacts of storms over Norway:

Wind-related damage claims account for 56% of Norway's insurance payouts related to natural hazards from 1980 to 2017 and are by far the largest component of loss claims related to natural hazards (DSB Norway, 2019). *They can affect all sectors from forest to marine and housing infrastructures (Gardiner et al., 2013; Jensen et al., 2010).* However, a detailed investigation into the relationship between Norwegian windstorms and damage has

so far not been conducted. The comparison of different proposed storm-damage models has only been conducted in a few countries due to a lack of long and sufficiently homogeneous insurance claims data (Cole et al., 2010; Prahl et al., 2015). *Determining the best storm-damage model is important in order to make accurate predictions of future damages, whether it be in a few days (short-term forecast) or in many years (climate change range).* In this paper, we investigate the relations between wind storms and their associated damage by analysing 36 years of daily insurance data on the municipality level and daily maximum wind speed data using a set of storm-damage functions. *Furthermore, we develop a probabilistic damage classifier that distinguishes between damaging and non-damaging wind speeds to help improve early warning systems.*

**Comment 3**

The review paper by Gliksman et al. (2023) provides a good overview on the topic and should therefore be included.

Thank you for bringing this new review paper to our notice. We have included the reference of this article in the second paragraph of the introduction*: Several methods in the literature assess the risk associated with extreme wind events across various sectors such as agriculture, transport, and energy at varying spatial resolutions (Gliksman et al., 2023). Storm-damage functions are one such method which describes the mathematical relation between the intensity of a natural hazard, here the wind speed, and the monetary loss incurred due to the event.*

**DATA**

**Comment 1**

Why did you choose the year 2015 to adjust the insurance loss for inflation?

Thanks for pointing this out. The official base year for the consumer price index (CPI) used by Statistics Norway is 2015 so we decided to follow them rather than recalculate the CPI values for another year. We have added the following text to explain*: To adjust for the effect of inflation, the insurance loss is adjusted using the Norwegian official consumer price index (CPI) at a fixed year (SSB Norway 2023a). The base year for CPI in Norway is 2015.*

**Comment 2**

Is the population data gridded (line 123) or at municipality level (line 95)?

The population data obtained from Statistics Norway as on line 123 is on a 1 km x 1 km grid and is available annually for the period 2001-2019. Therefore, we choose to use a constant population that is the average population for the period 2001-2019 in each grid cell. To get the population weighted wind speeds, we remapped the population field to the 3 km x 3 km NORA3 grid before taking the average of the population-weighted wind speeds in each

municipality as our "damage relevant" wind speed. However, the population data mentioned on line 95 in the original manuscript is also published by Statistics Norway but at the municipality level and is available for the whole period of the study. As the insurance data is also at the municipality level, we scale the insurance loss data with municipality-level population to get the loss per person. Hence both the insurance and the damage relevant wind speeds include information about the population. We have tried to make it clearer in the manuscript by adding the following lines: *Statistics Norway publishes yearly population data at municipality level which goes back to 1951 (SSB Norway, 2023b). To address the change in exposure to a certain extent, we compute the loss per person for each municipality by dividing the insurance loss data with the yearly population.*

Please see also our response to comment 1 in the Methods section.

**Comment 3**

You have adjusted the loss for inflation and then talk about the zero inflation of the loss time series. This is quite misleading.

We understand the confusion here. The inflation that we adjust for is financial inflation, while "zero inflation" is a statistical term that refers to the excess number of zeroes in the insurance loss data due to the absence of losses on most days. To add more clarity to this, we have added the following: *The presence of such extreme events brings skewness in the loss distribution and the absence of losses on most days of the year makes the loss data zero-inflated (excess number of zeros in the data).*

**Comment 4**

It is unclear which wind speed and gust data you extracted from NORA3:

- ☐ Line 113: time steps 4-9h
- ☐ Line 113: hourly wind speed
- ☐ Line 114: daily maximum near-surface wind speed.

We have changed the text to: *The hindcast consists of a sequence of 9-h forecasts initialised at 00, 06, 12 and 18 UTC every day from 1985 to 2020, which were the 36 years available at the time of our analysis. Aggregating the 4-9 h lead times provides an hourly dataset from which we extract the daily maximum wind speed and gust. A more comprehensive description can be found in the Appendix of Haakenstad et al. (2021).*

**METHODS**

**Comment 1**

It is not clear how the estimation of municipality level wind speed works.

As suggested by the reviewer, we have tried to clarify how we obtain the wind speed at the municipality level with the following text:

*As the insurance loss is at the municipality level, we must estimate a municipality-relevant wind speed to apply the storm-damage functions. A simple approach would be to use the daily average wind speed among all the grid points contained in a given municipality. However, to compensate for the complex topography and disparate demography of Norway, a more relevant population-weighted wind speed has been estimated to remove extreme wind events occurring over mountains, lakes, and other population-sparse regions. We weight the NORA3 daily maximum hourly wind speed with the gridded population. Statistics Norway publishes yearly gridded population data at 1 km x 1 km for Norway for the period 2001 to 2019 (https://www.ssb.no/natur-og-miljo/geodata; Strand and Bloch, 2009), which does not cover the whole period of the study. Therefore, we computed the average population for the period 2001-2019 in each grid cell. Then this averaged population is remapped on the same 3 km x 3 km grid as the NORA3 data. To achieve this, we assign each population grid cell to the nearest NORA3 grid cell. If more than one non-zero population grid cell corresponds to a NORA3 cell, we assign the sum of the population grid cells to the NORA3 grid cell. Finally, in order to have the wind speed at the municipality level, as is the insurance data, we take the population-weighted average of the daily maximum hourly wind speed in each municipality. We repeat the process for the daily maximum wind gusts.*

**Comment 2**

Why did you not calculate the storm damage by grid point and then aggregate it by region (municipality or national level), as Pinto et al. (2012) or Karremann et al. (2014a) did, for example? These studies also include a weighting with population density.

Using the Klawa damage function, Pinto et al. (2012) calculated the storm severity index at grid points and aggregated it to the storm affected region. Then they calibrated the aggregated loss indexes with observed insurance losses via linear regression (see eq. 2 by Pinto et al., 2012). In the present study, we follow a similar methodology, except that we chose to calibrate the Klawa damage damage function with insurance loss at municipality level. To our best knowledge, such an index based approach has never been performed yet at grid point level using other damage functions. Therefore, calibration of the damage functions has to be performed at the same resolution as the loss data, which, in our case, is at municipality level. Karremann et al. (2014a) follows the methodology of Pinto et al. (2012) but focuses on return periods of events and does not calibrate the loss indices with observed loss.

*The Klawa model was originally developed as a loss index for German districts and to estimate annual national losses using the German Insurance data. Later, Pinto et al. (2012) calibrated the damage function for the affected areas of individual storm events using the German insurance data. In the present study, we follow a similar methodology, except that we chose to calibrate the Klawa damage damage function with insurance loss at municipality level. Prahl et al. (2015) applied the damage function at district level on daily German insurance losses.*

**Comment 3**

Exponential model: Why did you chose the 95th percentile and not a similar threshold as in the other models? It seems too low to assess extreme events. Why is it necessary/useful to bin the loss with respect to wind speed?

Due to its shape, the exponential model can be extended to lower wind speeds that may cause small to medium damages. Therefore, we chose the 95th percentile of the wind speed to include lower wind speeds with the aim to get a model able to simulate losses for moderately strong wind speeds. The wind speed between the 95th and 98th percentiles makes up around 10% of total insured losses. In addition, the median number of loss days for municipalities is 64 for wind speeds above the 95th percentile and 49 for wind speeds above the 98th percentile. Thus, using the 95th percentile adds more data, for a better fit of the low loss events. The other model using a threshold is the Klawa model. This model is only suitable for extreme loss events and the 98th percentile is a more appropriate choice. We have added the following lines in section 2.4.1 for the justification of the choice of the threshold of the 95th percentile:

> The exponential model, by its shape, can be extended to lower wind speeds that may cause low to medium size losses. To take advantage of this, we choose the 95th percentile wind speed above which 82% of losses are recorded as the threshold for the exponential model. Such a threshold ensures that the model accounts for low to medium losses while discarding very small losses.

Losses are binned to increase the robustness of model fitting to events that are rare (large events). If the data is not binned the fitting procedure will minimise the deviation of all individual losses to the curve, leaving a good fit where there are many events (low losses) at the expense of the fit for the high losses. If the data is binned, equal weight is put on each bin, thereby increasing the goodness of fit for the extreme events at the expense of the fit for low events. Another advantage of binning loss with respect to wind speed is that the influence of one or two extreme losses on the shape of the damage function can be eliminated. We have clarified the purpose of binning the data in section 2.4:

> For robust storm-damage relations, extreme care should be taken when calibrating the damage functions. To ensure robustness of the damage functions, we bin the loss data with respect to wind speeds to reduce the weight of low loss events. Note that we do not perform binning for the Klawa damage function as the model is only suitable for high loss events and inherently removes the low losses with the use of a high wind speed threshold. More about binning in individual models is explained in the following sections.

**Comment 4**

Klawa model: Why do they use the 98th percentile? Please explain. Have you checked whether this threshold is suitable for Norway (see Karremann et al., 2014b)?

The 98th percentile wind speed threshold in the Klawa model is not particularly well justified in the literature. Klawa and Ulbrich (2003) used the 98th percentile wind speed as the threshold for two reasons: 1) the assumption by Palutikof and Skellern (1991) that storm damages occur in 2% of all days and 2) the German insurance threshold for storm damage claim settlement is 20 m/s which roughly coincides with the 98th percentile. For Norway, 72% of the insurance losses are caused by wind speed above the 98th percentile. Given that the Klawa model only is suitable for high loss cases, the 98th percentile seems like a reasonable choice. We agree that rather than a fixed deterministic threshold, statistically-determined estimates for wind speed thresholds are desired, but it is not clear how this could be best done. Thus. for simplicity, we have chosen not to do this. One alternative would be to make the threshold municipality dependent. Another alternative would be to have a fixed value as the threshold. The fixed threshold approach was used in Karremann et al. (2014b) assuming a threshold of 9 m/s for wind speeds causing damage in Norway. From our analysis, 75% of the municipalities exhibit a 98th percentile population-weighted wind speed above 9 m/s. Thus, our threshold is higher than in Karremann et al. (2014b). We have added the following paragraph in section 2.4.2 for the justification of choice of wind speed threshold:

*Several studies across Europe use the 98th percentile wind speed as a threshold for the Klawa damage function (Pinto et al., 2012; Karremann et al., 2014a, b). Ideally, the threshold for damaging wind should be locally chosen using statistically-determined estimates, but for simplicity we have kept the often used 98th percentile. In Norway, 72% of the insurance losses are caused by wind speeds above the 98th percentile. As the Klawa model is not designed for low loss cases, this is a fairly reasonable simplification. Note that if grid point wind speeds were chosen, this choice of percentile can be problematic for places with weak winds, such as southeastern Norway (see Fig. 7a). Therefore, for example, Karremann et al. (2014b) and Little et al. (2023) suggested a fixed 9 m/s as threshold for wind speeds causing damage in Norway. This study uses the mean population weighted winds speeds, reducing the relative importance of grid cells with very low wind speeds and therefore avoiding the problem of very low 98th percentile wind speeds.*

**Comment 5**

Why are you interested in different loss classes? Losses are primarily caused by gusts above a certain threshold.

Society is primarily interested in events with very high losses. Thus, investigating the quality of the damage functions for these events is of particular interest. The extreme loss class we have defined here comprises, on average, one loss day per year but accounts for 85% of the total losses. The high loss class, i.e. losses that lie between the 98 and 99.7th loss

percentiles, comprises only 8% of the total losses. If we decided not to segregate into loss classes, the validation would be governed by the numerous low-loss events whereas it is clearly the high-loss events that are of most interest. The loss classes therefore help us to differentiate the quality of the damage functions for events of different severity. We will better justify the use of loss classes in the manuscript in section 3.1 with the following lines:

From the damage perspective, extreme damaging events are of the topmost concern. The losses above the 99.7th percentile (occurring on average around one time each year) in each municipality when aggregated account for 85% of total national loss. In each municipality we define the losses higher than the 99.7th percentile as the extreme loss class and losses lying between the 98th and 99.7th percentile as the high loss class. *The high loss class comprises 8% of losses.* The extreme loss class represents approximately 31 and 9 extreme loss days in each municipality in the training and testing data, respectively. *Segregation of losses into different classes helps to assess the performance of the damage functions for events of different severity.*

**Comment 6**

Modified Prahl model: Why did you choose this particular modification?

Initial analysis showed that the power law-based probabilistic damage function by Prahl underestimated high losses in many municipalities. At the same time, the deterministic-based exponential models were showing good fits. Therefore, the modified Prahl is an attempt to keep the probabilistic aspect of the Prahl model, in combination with an exponential fit. We have added the following lines to justify our modification:

*The rationale behind Prahl's damage function is that the loss increases steeply for extreme wind events (Fig. 2c). However, based on inspection of the quality of the fitted curves for very high loss events, we identified a need for an even steeper damage function for certain municipalities in Norway. The deterministic exponential damage function increases sharply and shows good fits for municipalities in Norway. Therefore, we propose a modified version of the damage function by Prahl that combines an exponential fit with the probabilistic aspect of the Prahl model.*

**Comment 7**

The concept of the damage classifier is not clear. What added value does it offer compared to the exponential or Klawa model, which already classify events/non-events based on wind speed thresholds?

A damage classifier classifies a given wind speed as damaging or not. While the exponential and Klawa models are deterministic models without any information about damage probabilities for different wind speeds, we can extract from the probability term in the Prahl damage function the probabilities that a wind speed causes damage. For our damage classifier, we determine the probability threshold from the damage probabilities.

To demonstrate the advantage of a probabilistic damage classifier, we can compare the accuracy of the fixed wind speed threshold used in the Klawa model to the accuracy of the probabilistic damage classifier based on the Prahl damage function. For extreme wind speeds, i.e for wind speeds above the 98th percentile, the accuracy (proportion of correct prediction) of the wind speed threshold based classifier is far lower (median of 18%) (Fig. R1a below) than the accuracy of the probabilistic damage classifier (median of 68%) (Fig. R1b below). This can be explained by the fact that, while the Klawa model will always estimate damage above a given wind threshold (98th percentile) and no damage below, the damage classifier provides a damage/no damage classification based on the historical probability of damage for any given wind speed.

[Figure]

**Figure R1**: For windspeeds above the 98th percentile, histograms of (a) accuracy of damage classifier based on 98th percentile windspeed (median: 18%) and (b) accuracy of damage classifier   defined from the Prahl damage function (median: 68%)

To demonstrate the usefulness of the damage classifier we added the following paragraph in section 3.5:

*The damage classifier defined from event occurrence probabilities clearly outperforms classifiers that solely rely on wind speed. To demonstrate this, we define a damage classifier based on wind speed thresholds in which all wind speeds above the 98th percentile are labelled as damaging (as is done in the Klawa model). For wind speeds above the 98th percentile, the damage classifier based on the probability term of Prahl damage function (eq. 4) shows far higher accuracy when compared to the accuracy of the classifier solely based on wind speed (Fig. R1).*

**Comment 8**

If you focus on daily losses (line 238), how do you account for longer lasting events? Which day of the storm event is selected as the 'loss day'?

Wind-related loss events very seldom exceed a full day within a given municipality. If that is the case and a long event has led to losses every day of the event, every day  is  treated as an individual event. We have added this precision in the manuscript.

**FIGURES / TABLES**

**Comment 1**

In total, 14 figures and 3 tables are too many to include in the main manuscript. You should select the most relevant ones to convey your key findings and move the rest to the Supplementary Material.

Following the reviewer's suggestion, we have moved Figs. 1, 5, 6, 7, 13 and 14 to the supplement. We have also combined Figs. 2 and 3, Fig. 8 and Fig. 9. We have also moved Table 1 and Table 3 to the supplement. We have, however, added Figure R1 in section 3.5 (see our reply to comment 7 in methods). After these changes, the manuscript contains seven figures and one table.

**Comment 2**

In your spatial maps, you use the same colours for loss, wind speed, errors, thresholds, etc. Consider using different colormaps to depict the spatial patterns. This could help the reader better understand what is shown and clearly differentiate the different type of plots.

Thanks for the suggestion. We will change the colorbars in the revised manuscript.

**References**

Cole, C. R., Macpherson, D. A., and McCullough, K. A.: A comparison of hurricane loss models, Journal of Insurance Issues, 33, 31–53,http://www.jstor.org/stable/41946301, 2010.

DSB Norway: Analyses of Crisis Scenarios 2019, https://www.dsb.no/globalassets/dokumenter/rapporter/p2001636_aks_2019_eng.pdf, 2019.

Gardiner, B., Schuck, A. R. T., Schelhaas, M.-J., Orazio, C., Blennow, K., and Nicoll, B.: Living with storm damage to forests, vol. 3, European Forest Institute Joensuu, https://efi.int/sites/default/files/files/publication-bank/2018/efi_wsctu3_2013.pdf, 2013.

Gliksman, D., Averbeck, P., Becker, N., Gardiner, B., Goldberg, V., Grieger, J., Handorf, D., Haustein, K., Karwat, A., Knutzen, F., Lentink, H. S., Lorenz, R., Niermann, D., Pinto, J. G., Queck, R., Ziemann, A., and Franzke, C. L. E.: Review article: A European perspective on wind and storm damage – from the meteorological background to index-based approaches to assess impacts, Natural Hazards and Earth System Sciences, 23, 2171–2201, https://doi.org/10.5194/nhess-23-2171-2023, 2023

Haakenstad, H., Breivik, Ø., Furevik, B. R., Reistad, M., Bohlinger, P., and Aarnes, O. J.: NORA3: A Nonhydrostatic high-resolution hindcast of the North Sea, the Norwegian Sea, and the Barents Sea, J. Appl. Meteor., 60, 1443–1464, https://doi.org/10.1175/JAMC-D-21-0029.1, 2021.

Jensen, Ø., Dempster, T., Thorstad, E., Uglem, I., and Fredheim, A.: Escapes of fishes from Norwegian sea-cage aquaculture: causes, consequences and prevention, Aquaculture Environment Interactions, 1, 71–83, https://doi.org/10.3354/aei00008, 2010.

Karremann, M., Pinto, J., Von Bomhard, P., and Klawa, M.: On the clustering of winter storm loss events over Germany, Natural Hazards and Earth System Sciences, 14, 2041–2052, https://doi.org/10.5194/nhess-14-2041-2014, 2014a.

Karremann, M. K., Pinto, J. G., Reyers, M., and Klawa, M.: Return periods of losses associated with European windstorm series in a changing climate, Environmental Research Letters, 9, 124 016, https://doi.org/10.1088/1748-9326/9/12/124016, 2014b.

Klawa, M. and Ulbrich, U.: A model for the estimation of storm losses and the identification of severe winter storms in Germany, Nat. Hazards Earth Syst. Sci., 3, 725–732, https://doi.org/10.5194/nhess-3-725-2003, 2003.

Little, A. S., Priestley, M. D., and Catto, J. L.: Future increased risk from extratropical windstorms in northern Europe, Nature Communications, 14, 4434, https://doi.org/10.1038/s41467-023-40102-6, 2023.

Palutikof, J. P. and Skellern, A. R.: Storm Severity over Britain, A Report to Commercial Union General Insurance, Climatic Research Unit, School of Environmental Sciences, University of East Anglia, Norwich (UK), 1991.

Pinto, J. G., Karremann, M. K., Born, K., Della-Marta, P. M., and Klawa, M.: Loss potentials associated with European windstorms under future climate conditions, Climate Research, 54, 1–20, https://doi.org/10.3354/cr01111, 2012.

Prahl, B. F., Rybski, D., Burghoff, O., and Kropp, J. P.: Comparison of storm damage functions and their performance, Nat. Hazards Earth Syst. Sci., 15, 769–788, https://doi.org/10.5194/nhess-15-769-2015, 2015.

SSB Norway: Consumer price index, https://www.ssb.no/en/priser-og-prisindekser/konsumpriser/statistikk/konsumprisindeksen, 2023a.

SSB Norway: Population 1 January and population changes during the calendar year (M) 1951 - 2023, https://www.ssb.no/en/statbank/table/06913/, 2023b.

Strand, G.-H. and Bloch, V. H.: Statistical grids for Norway, 9, https://www.ssb.no/a/english/publikasjoner/pdf/doc_200909_en/doc_200909_en.pdf, 2009.

---

## Author Comment (AC2)

**Reply to referee comment 2**

**Windstorm damage relations - Assessment of storm damage functions in complex terrain**

We thank the reviewer for reading our manuscript and giving thoughtful comments and suggestions. A detailed response to all comments is found below in blue.

This paper compares the performance of four different damage functions in estimating windstorm losses over Norway. The damage functions are assessed at municipality and national spatial scales, and daily and annual time scales. All damage functions can reproduce the spatial loss patterns of the most extreme storms, and when aggregated nationally have high temporal correlations with observations on daily timescales. Time series of national aggregate losses on annual timescales correlate well with the deterministic damage models, but the probabilistic models produce very large errors for some years. Using the probability of damage function from the probabilistic models, a damage classifier is developed to distinguish between damaging and non-damaging wind speeds at the municipality level. The classifier has a low hit rate when considering all events, but does predict the most extreme events.

I think this is a worthwhile study and the paper has some interesting results. However, I have a few major concerns that need addressing before I can recommend publication. Detailed comments are listed below.

**Major comments**

**Comment 1**

I think it needs to be emphasised that the damage functions were developed to work on different spatial and temporal scales. For example, the Klawa and Ulbrich (2003) cubic model was originally used for annually and nationally aggregated data, whereas the Prahl et al (2012) model was applied to smaller scale daily data (hence the need to be stochastic). It is surprising that the cubic model works so well on the municipality scale.

We agree with the reviewers' observation on the need to emphasise the characteristics of the damage functions. As you pointed out, initially, the Klawa and Ulbrich (2003) cubic model was applied to the annual and national aggregated data. Donat et al. (2011) and Pinto et al. (2012) refined the cubic model and applied it to the district level insurance loss data. Later, Prahl et al. (2015) applied several models, including the cubic model to daily insurance losses for the districts in Germany and found that the application of the cubic model should be restricted to model extreme loss events. We have added the following in section 2.4.2: *The Klawa model was originally developed as a loss index for German districts and to estimate annual national losses using the German Insurance data. Later, Pinto et al. (2012) calibrated the damage function for the affected areas of individual storm events using the German insurance data. In the present study, we follow a similar methodology, except that we chose to calibrate the Klawa*

*damage damage function with insurance loss at municipality level. Prahl et al. (2015) applied the damage function at district level on daily German insurance losses.*

**Comment 2**

As you mention there are zeros in the daily municipality data, so for the linear regression step for the exponential and cubic models ($L(v) = \beta 0 + \beta 1 d(v)$) the residuals will not be Gaussian. It's not clear to me how you dealt with the zeros – were they included in the fit? Was the data binned for the cubic model as well as the exponential one?

While fitting the damage functions, we focused on the extreme observations and tried to minimise the impact of zero and low losses. For the exponential model, the losses are split into bins and at least five loss days (i.e non-zero losses) should belong to each bin. The zero losses in the bins that satisfy the described conditions are also used to obtain the bin average loss. The bin average losses are then log-transformed to fit the exponential damage function. For the Klawa model, we use all the loss data above the 98th percentile wind speed (including zeroes) and do not apply binning. We do not perform binning for the Klawa damage function as the model is only suitable for high loss events and inherently removes the low losses with the use of a high wind speed threshold.

**Comment 3**

For the cubic model, in this paper each municipality is fitted separately, whereas in the Klawa & Ulbrich (2003) they aggregate the data nationally (and annually) then apply the linear regression fit. How much does the regression fit vary between municipalities? i.e. did you prove it's necessary to fit each region separately?

Klawa and Ulbrich (2003) used annual insurance data at national level and therefore could not fit storm-damage functions at a smaller level, such as region or municipality level. However, we have insurance data at higher spatial resolution and took advantage of it by fitting the storm-damage functions at the municipality level. Moreover, using municipality level is more relevant to forecasters who will have information on potential damage at such spatial level that they can issue a warning for a local region (and not nationally). We have added this justification of our method in the manuscript.

**Comment 4**

Klawa & Ulbrich (2003) only fitted above the 98th percentile assuming that no damage occurred below this. How good an approximation is this for the Norwegian data? From fig 1 there is clearly damage below the 98th percentile but hard to tell how significant this is because I assume there is a very high proportion of days with zero loss at lower wind speeds.

The 98th percentile wind speed threshold in the Klawa model is not particularly well justified in the literature. Klawa and Ulbrich (2003) used the 98th percentile wind speed as the threshold for two reasons: 1) the assumption by Palutikof and Skellern (1991) that storm damages occur in 2% of all days and 2) the German insurance threshold for storm damage

claim settlement is 20 m/s which roughly coincides with the 98th percentile. For Norway, 72% of the insurance losses are caused by wind speed above the 98th percentile. Given that the Klawa model only is suitable for high loss cases, the 98th percentile seems like a reasonable choice. We agree that rather than a fixed deterministic threshold, statistically-determined estimates for wind speed thresholds are desired, but it is not clear how this could be best done. Thus, for simplicity we have chosen not to do this. One alternative would be to make the threshold municipality dependent. Another alternative would be to have a fixed value as the threshold. Karremann et al. (2014b) assumed a minimum threshold of 9 m/s for wind speeds causing damage in Norway. From our analysis, 75% of the municipalities exhibit a 98th percentile population-weighted wind speed above 9 m/s. Thus, our threshold is higher than previous assumed thresholds. We have added the following paragraph in section 2.4.2 for the justification of choice of wind speed threshold:

*Several studies across Europe use the 98th percentile wind speed as a threshold for the Klawa damage function (Pinto et al., 2012; Karremann et al., 2014a, b). Ideally, the threshold for damaging wind should be locally chosen using statistically-determined estimates, but, for simplicity, we have kept the often used 98th percentile. In Norway, 72% of the insurance losses are caused by wind speeds above the 98th percentile. As the Klawa model is not designed for low loss cases, this is a fairly reasonable simplification. Note that if grid point wind speeds are chosen, this choice of percentile can be problematic for places with weak winds, such as southeastern Norway (see Fig. 7a). Therefore, for example, Karremann et al. (2014b) and Little et al. (2023) suggested a fixed 9 m/s as threshold for wind speeds causing damage in Norway. The present study uses the mean population weighted winds speeds reducing the relative importance of grid cells with very low wind speeds and therefore avoiding the problem of very low 98th percentile wind speeds.*

**Comment 5**

L214: It's confusing to define the evaluation of damage classifier here, before you've defined the damage classifier. Maybe put section 2.6 before 2.5?

Thanks for the suggestion. Since defining a damage classifier involves metrics such as precision and recall, we will also define them along with the methodology of the damage classifier.

**Comment 6**

L256 and L326: "The choice of wind data has the potential to influence the performance of the damage functions…" and "The predictive performances of the damage function and the damage classifier confirms the importance of weighting wind speed with population for better performance of the damage functions." In the paper you show that the population weighting gives different maximum wind speeds for the municipalities (Fig 7b), but you don't actually show that it makes the damage models perform better. Does it?

Thank you for your comment. To address it, we have fitted the damage functions using the raw wind speeds instead of the population-weighted wind speeds. Figure R1 here below shows that the population weighted wind speed has lower coefficient of variance (CV) in 67% of the municipalities (including almost all of the high population municipalities) than the raw wind speeds, hence highlighting the usefulness of the population-weighting step in the majority of the municipalities. We have added the following paragraph in section 3.2:

*To demonstrate the advantage of weighting wind speed with population, damage functions were also fitted with the original wind speed as the predictor variable. The prediction error on the test data shows that the population weighted wind speed has lower CV in 67% of municipalities (see also Fig. R1 for the spatial distribution of where each wind speed input data performs better). From these results, we conclude that weighting wind speeds with population improves the predictive performance of the damage functions. Therefore, from now on, we only use the population weighted wind speeds when fitting the damage functions.*

[Figure]

Figure R1: Municipalities with smallest CV among the five models with weighted and maximum wind speeds on test data. Municipalities in grey are where maximum wind speed shows lower and blues are weighted wind speed shows lower CV.

**Comment 7**

In the abstract and conclusions (L324) it's stated that all models perform well on national scales, but from Fig S4 it looks like the probabilistic models perform very poorly on this scale.

The reviewer is correct, the probabilistic models significantly overestimate the losses in certain municipalities, as reflected in Fig. S4. However, if the 15-20% municipalities that have poor fit are removed from the estimation, the probabilistic models are also able to reproduce the annual national losses. We have tried to make it clearer in the abstract and conclusion.

We removed 'annual national losses' in line 324 and rephrased to: *The models' ability to reproduce spatial loss patterns of extreme loss events with a high degree of accuracy confirms the utility of both deterministic and probabilistic damage functions in estimating extreme loss events.*

In the abstract, the sentence *The good agreement between the observed and estimated losses at municipality and national levels suggests that the damage functions used in this study are well suited for estimating severe wind storm-induced damages.* has been rephrased to *There is no single damage function that outperforms others. However, a good agreement between the observed and estimated losses at municipality and national levels for a combination of damage functions suggests their usability in estimating severe wind storm-induced damages.*

**Minor comments:**

**Comment 1**

L103 "Figure 1 highlights a record high number of claims in years 1992, 2011 and 2015. This can be attributed to the New Year Storm in 1992, storm Dagmar in 2011 and storms Nina and Ole in 2015 (Table 1)." In the figure it looks like there's high loss in 1994, not 1992.

Thank you for your careful reading, we have changed the text to: *Figure 1 highlights a record high number of claims in years 1994, 2011 and 2015. This can be attributed to the Storm of 1994, storm Dagmar in 2011 and storms Nina and Ole in 2015 (Table S1).*

**Comment 2**

Fig 2: How do you have zero on a log scale (y-axis)? It looks like zero losses are not actually plotted so this should be stated.

The reviewer is correct. The following line is added to the caption of Fig 2: *Note that the y-axis is on a logarithmic scale and the zero loss on the y-axis is only for reference but the zero losses are not plotted/displayed.*

**Comment 3**

Section 2.5, L204 "For robust storm-damage relations, extreme care should be taken in the parameters estimation of damage functions. To ensure robustness of the damage functions, we bin the loss data with respect to wind speeds to eliminate the sensitivity of damage functions to extreme events." I'm not sure what you mean by this. Are you talking about the binning done when fitting the parameters, or do you bin the data when evaluating the errors as well?

We understand the lack of clarity here. We only bin the losses when fitting parameters. The above mentioned lines are deleted and the following is added in section 2.4:

*For robust storm-damage relations, extreme care should be taken when calibrating the damage functions. To ensure robustness of the damage functions, we bin the loss data with respect to wind speeds to reduce the weight of low loss events. Note that we do not perform binning for the Klawa damage function as the model is only suitable for high loss events and inherently removes the low losses with the use of a high wind speed threshold. More about binning in individual models is explained in the following sections.*

**Comment 4**

Section 3.3: Since storms last more than one day, how did you estimate the losses for a single storm? e.g. did you sum the losses over a few days? Choose the maximum loss day?

For the major recorded historical storms, we sum the loss days given in Table 1. We have added the following lines in the first paragraph of section 3.3: *To compare the estimated and observed losses caused by major storm events, we sum the losses within the date range given by the Norwegian Natural Perils Pool and written in Table 1.*

To make it clear we have made the following changes.

We add the following to the title of Table 1: *The event periods are as defined by the Norwegian Natural Perils Pool.*

And in the caption of Fig. 10 and Fig. S2: *For each storm, we sum all the loss days as given in Table 1.*

**Comment 5**

Fig 12 caption – what does it mean 'Annual time series of observed and estimated national losses using the extreme loss class'? Are these not just annually aggregated losses (i.e. the sum of all days?)

Fig. 12 is the annual aggregate loss in the extreme loss class. We have modified the caption accordingly: *Annually aggregated national losses using only the loss days in the extreme loss class from the insurance data (red line) along with the annual national loss estimates (blue line),*

*which are the sum of each municipality's best-performing-model estimate (see also Table 2). Note that the y-axis is logarithmic and the shaded region represents the testing period.*

**Comment 6**

Fig 11 – panels aren't labelled.

Thanks for bringing this to our attention. We will label the panels in the revised manuscript.

**References**

Donat, M. G., Leckebusch, G. C., Wild, S., and Ulbrich, U.: Future changes in European winter storms losses and extreme wind speeds inferred from GCM and RCM multi-model simulations, Nat. Hazards Earth Syst. Sci., 11, 1351–1370, https://doi.org/10.5194/nhess-11-1351-2011, 2011.

Karremann, M., Pinto, J., Von Bomhard, P., and Klawa, M.: On the clustering of winter storm loss events over Germany, Natural Hazards and Earth System Sciences, 14, 2041–2052, https://doi.org/10.5194/nhess-14-2041-2014, 2014a.

Karremann, M. K., Pinto, J. G., Reyers, M., and Klawa, M.: Return periods of losses associated with European windstorm series in a changing climate, Environmental Research Letters, 9, 124 016, https://doi.org/10.1088/1748-9326/9/12/124016, 2014b.

Klawa, M. and Ulbrich, U.: A model for the estimation of storm losses and the identification of severe winter storms in Germany, Nat. Hazards Earth Syst. Sci., 3, 725–732, https://doi.org/10.5194/nhess-3-725-2003, 2003.

Little, A. S., Priestley, M. D., and Catto, J. L.: Future increased risk from extratropical windstorms in northern Europe, Nature Communications, 14, 4434, https://doi.org/10.1038/s41467-023-40102-6, 2023.

Palutikof, J. P. and Skellern, A. R.: Storm Severity over Britain, A Report to Commercial Union General Insurance, Climatic Research Unit, School of Environmental Sciences, University of East Anglia, Norwich (UK), 1991.

Pinto, J. G., Karremann, M. K., Born, K., Della-Marta, P. M., and Klawa, M.: Loss potentials associated with European windstorms under future climate conditions, Climate Research, 54, 1–20, https://doi.org/10.3354/cr01111, 2012.

Prahl, B., Rybski, D., Kropp, J., Burghoff, O., and Held, H.: Applying stochastic small-scale damage functions to German winter storms, Geophys. Res. Lett., 39, https://doi.org/10.1029/2012GL050961, 2012

Prahl, B. F., Rybski, D., Burghoff, O., and Kropp, J. P.: Comparison of storm damage functions and their performance, Nat. Hazards Earth Syst. Sci., 15, 769–788, https://doi.org/10.5194/nhess-15-769-2015, 2015.